# Principle and Application of Steam Explosion Technology in Modification of Food Fiber

**DOI:** 10.3390/foods11213370

**Published:** 2022-10-26

**Authors:** Chao Ma, Liying Ni, Zebin Guo, Hongliang Zeng, Maoyu Wu, Ming Zhang, Baodong Zheng

**Affiliations:** 1Department of Food Science, Fujian Agriculture and Forestry University, Fuzhou 350002, China; 2Jinan Fruit Research Institute All China Federation of Supply and Marketing Co-Operatives, Jinan 250014, China

**Keywords:** steam explosion, dietary fiber, modification, promote dissolution, application

## Abstract

Steam explosion is a widely used hydrothermal pretreatment method, also known as autohydrolysis, which has become a popular pretreatment method due to its lower energy consumption and lower chemical usage. In this review, we summarized the technical principle of steam explosion, and its definition, modification and application in dietary fiber, which have been explored by researchers in recent years. The principle and application of steam explosion technology in the modification of food dietary fiber were analyzed. The change in dietary fiber structure; physical, chemical, and functional characteristics; the advantages and disadvantages of the method; and future development trends were discussed, with the aim to strengthen the economic value and utilization of plants with high dietary fiber content and their byproducts.

## 1. Introduction

With the development of living and economic standards, people have placed more importance on healthy diet requirements by eating more plant-based food like fruits and vegetables, which helps to maintain a healthy lifestyle. Cereals, fruits, and vegetables and other plant-derived foods are rich in functional compounds such as vitamins and dietary fiber. However, at present, more than one-third of the fruits and vegetables and their processing byproducts are not fully processed and utilized. These plant resources are usually treated by animal feed, landfill or incineration, and producers usually pay a certain price for it [1,2]. Thus, this inefficient processing of plant-derived food potentially has negative impacts on the environment.

Dietary fiber (DF) is a class of carbohydrate polymers that are neither digested nor absorbed in the small intestine, comprised of cellulose, hemicellulose, pectin, algae, and lignin. Dietary fiber mainly exists in the tough wall layer of plant cells. According to relative water solubility, dietary fiber can be divided into soluble and insoluble forms. These substances make up the hulls of cereal and wheat; and the roots, skin, stems and leaves of vegetables and fruits, which play an important beneficial physiological role. Over the years, research regarding the potential health benefits of dietary fiber in disease prevention has received considerable attention; obesity, type II diabetes, and cardiovascular diseases have been extensively studied [3]. Many previous studies found that there is an inverse relationship between dietary fiber intake and change in body weight. Koh-Banerjee et al. [4] supported this statement in a study that for every 40 g/d increase in whole grain intake, weight gain decreased by 1.1 lbs. A majority of studies show a positive correlation between high glycemic foods and type II diabetes. However, Meyer et al. [5] found that there was a strong inverse relationship between dietary fiber intake and diabetes. In the study, women who consumed an average of 26 g/d of dietary fiber had a 22% decreased risk of getting diabetes than women who consumed just 13 g/d. Recent studies suggest that increasing levels of dietary fiber may improve carbohydrate metabolism in a non–pharmacological way, resultingly having a positive effect on weight control and diabetes prevention. High quality DF should include more than 10% SDF and have favorable processing properties, as well as physiological activity and a healthcare function. Although the amount of total dietary fiber is fairly high in many plants, the content of SDF is only approximately 3–4% of the total dietary fiber. Insoluble dietary fiber may have some outstanding problems, such as rough taste, poor water-holding capacity, poor swelling power and weak functional activity; consequently, the utilization of this kind of raw material is not high [6]. It is urgent to carry out cost-effective industrial treatment of these resources and transform as much insoluble dietary fiber into soluble fiber as possible, which means modifying the dietary fiber raw materials in an efficient green way, in order to make better use of these resources.

As for the high-value utilization of raw materials rich in dietary fiber, pretreatment is needed to destroy their relatively dense physical structural barrier, so as to promote the efficient extraction, transformation, and utilization of DF [7]. At present, dietary fiber modification strategies may be classified into four types: physical, chemical, biological and combined method [7]. The glycosidic bond of DF is melted or broken using a physical method that makes use of high temperature, high pressure, immediate pressure decrease, explosion, high-speed impact, and shearing. This accomplishes the goal of modification, such as in steam explosion (SE) [8], ultrafine comminution (UC) [9], ultrasound [10], microwave [11], etc. The chemical method alters the structural and functional characteristics of DF by chemical reactions, such as alkaline hydrogen peroxide (AHP) treatment, alkali treatment, acid treatment, and Na_2_HPO_4_ treatment [12,13]. Compared with the physical method, the chemical method shows a short processing time and can react at room temperature. However, after the chemical method treatment, the modified DF exhibited low purity, and was prone to produce harmful components [14]. Utilizing certain enzymes or microorganisms to enzymatically hydrolyze or ferment raw materials with the goal of changing the content and bioactivity of DF is known as the biological method [15,16]. Biological methods are widely used in the modification of DF due to their high specificity, and benefits of milder processing conditions and environmental friendliness. The disadvantages of biological methods might be the high cost of enzyme purification and strain breeding [7]. Combining two or more ways to modify the DF is referred to as a combined modification method. Since the biological method requires a mild environment, the conditions of chemical methods are very harsh. Overall, the physical method is the main choice to combine with the other three methods [17,18]. Among these modification methods of DF, the physical methods are frequently used in the modification of dietary fiber because their advantages of low cost, short time consumption, simple operation, and lack of toxic waste generation [7].

Steam explosion (SE) technology, as a new type of physicochemical modification technology of food raw materials, has increasingly been studied by more and more researchers in recent years. SE is a method that presses high-pressure and high-temperature steam into cell walls and plant tissues, applying the thermochemical action of high-temperature cooking coupled with the physical tearing action of instantaneous blasting [8,19,20]. The physicochemical properties of the macromolecules of the fibrous raw materials are changed, thereby promoting the subsequent separation and conversion of solid-phase multi-component materials, which entails heating the biomass under pressurized steam to explode the cellulose fibrils and depolymerize lignin. At present, the steam explosion pretreatment has been used for ethanol production from straw materials, such as switchgrass and sugarcane bagasse [21], corn stover [22] and sunflower stalks [23]. In addition, the wall-breaking effect of SE on natural products has been widely studied. To sum up, this method has been gradually applied to the processing and modification of dietary fiber raw materials due to its low energy consumption and chemical usage [24,25].

In this review, we summarize the principle of structural modification of dietary fiber by steam explosion technology explored by researchers in recent years. The change in dietary fiber structure; physical, chemical, and functional characteristics; advantages and disadvantages of the method; and future development trends are discussed, with the aim to strengthen the economic value and utilization of plants with high dietary fiber content and their byproducts.

## 2. Source and Classification of Dietary Fiber in Food

In the 1950s, Eben Hipsley [26] first proposed the definition of dietary fiber (DF), which described the food components from the cell walls of plants, and many scholars have since carried out research on DF. As suggested by Trowell et al. [27], DF was characterized as lignin-resistant and plant polysaccharides to hydrolyze human digestive enzymes. The American Society of Cereal Chemists includes oligosaccharides (DP 3–9) in the DF definition, as these substances have some physiological characteristics with the majority of DF. According to the AACC, DF is the edible section of plants, or carbohydrate-like substances, that are difficult for humans to digest and absorb in the small intestine; but are fully or partially fermented in the large intestine [28]. Polysaccharides, lignin, and associated plant matter are all included in DF. This was the first time that DF-related substances, such as phenolic compounds, were included in the definition of DF [28]. In 2009, the Codex Alimentarius Commission defined DF as 10 or more monomeric units of carbohydrate polymers (whether to include 3 or 9 monomer chain carbohydrates is determined by the national authority) that are not hydrolyzed by the endogenous enzymes in the human small intestine and fall into one of the following categories: (1) edible carbohydrate polymers naturally occurring in the food as consumed; (2) synthetic carbohydrate polymers or carbohydrate polymers that have been obtained from food raw material by physical, enzymatic, or chemical means and which have been shown to have a physiological effect or benefit to health as demonstrated by generally accepted scientific evidence to competent authorities [29]. With the development of nutrition and other disciplines, dietary fiber (DF) is known as the seventh most important nutrient by the nutrition community, and plays an important role in human health. DF exists in most natural plants, such as fruits (16.74–91.24%), vegetables (6.53–85.19%), grains (9.76–69.20%) and so on [30]. Indeed, dietary fiber is a heterogeneous complex of components with different physical, chemical, and physiological properties, which complicates direct analytical measurements [31]. DF is an important component of a healthy diet; the positive correlation with human health has been established by the scientific community [32]. More than 50% of functional foods in the market contain DF as the active ingredient [33]. Additionally, DF has many biologically active substances that are beneficial for our health, such as improving the intestinal flora, lowering blood glucose, decreasing the probability of obesity and cardiovascular disease, reducing the risk of some cancers, increasing fecal volume, promoting bowel movements, etc. [26,34,35,36,37].

The Chinese Nutrition Society defines DF as a carbohydrate polymer that is not easily digested by digestive enzymes, mainly from the plant cell wall, and comprised of cellulose, hemicelluloses, lignin, pectin and resin; more specifically defined as follows: (1) Cellulose is a long-chain polymer composed of glucose linked by β-1,4 glycosidic bonds; (2) Hemicellulose is a polymer composed of a mixture of monosaccharides such as arabinose, galactose, and xylose; (3) Lignin is not a polysaccharide, but a polymer composed of phenylpropane units; (4) Pectin is a polymer composed of uronic acid residues with rhamnose and contains neutral sugar branched chains; (5) Mucus and gums are mostly hemicelluloses. The existence of bioactive substances linked to the cell wall has a significant impact on the physicochemical characteristics of DF and also affects its physiological properties in humans. Additionally, according to the DF definition supplied by the AACC, phenolic compounds were included in the definition of DF. These components may play a significant role in DF properties, even though the properties are generally attributed to those of polysaccharides [33]. Goñi et al. [38] found that polyphenols appear in both fractions of fiber, but in a greater proportion in the insoluble portion. Consequently, polyphenols are an essential component of DF, and the definition of DF restricted to non-digestible polysaccharides and lignin could be extended to include polyphenols [39]. According to the solution properties of DF, they can be divided in two types: insoluble dietary fiber (IDF) and soluble dietary fiber (SDF) [40]. Additionally, the degree of fermentation in the colon can be divided into complete fermentation fibers (mostly SDF) and partial fermentation fibers (mostly IDF). SDF is mainly fermented by bacteria in the ileum and ascending colon, while IDF is primarily fermented in the distal colon. Many kinds of natural DF tend to have high IDF content and low SDF content, resulting in the rough taste of raw materials; poor functional properties such as water-holding capacity (WHC), oil-holding capacity (OHC), and swelling capacity (SC); and low physiological activity, which have seriously limited the development and utilization of dietary fiber resources [7]. Thus, finding the most appropriate modification method to enhance the yield and functional properties of DF is becoming the hot topic in the food processing research fields.

## 3. DF Modification Methods

The current DF modification methods can be divided into four types: the physical method, chemical method, biological method, and combined method [7]. The above various modification methods have their own advantages and disadvantages. Due to their low cost, straightforward operation, and lack of harmful waste formation, physical methods are frequently used in the modification of DF. But, some physical methods require a large operating area and involve high levels of danger. Compared with physical and biological methods, chemical methods can create DF quickly and at room temperature, but they also have a tendency to produce a lot of hazardous byproducts. The products prepared by the biological method have the advantages of high purity and fewer byproducts. However, the only disadvantage of the biological method may be the high cost of enzymes and long operation cycles [7]. Based on the advantages and disadvantages of the above methods, it is urgent to find a DF modification method with low cost, high efficiency, a short reaction time, lower toxicity, fewer harmful byproducts, large processing capacity, and easy industrialization.

## 4. Steam Explosion Technology Process and Principle

Steam explosion (SE), also known as autohydrolysis, is an atypical hydrothermal pretreatment. According to the SE principle, the raw materials are placed into a cylinder at a high temperature and high pressure. The steam that is generated then permeates into the interior of the materials and fills the tissue pores with steam. Finally, the high pressure created by the saturated steam is immediately released within milliseconds [41]. The water contained in the substrate evaporates and rapidly expands, which causes cell wall rupture to form pores, so that small molecular material is released from the cells [42]. This also causes the reduction of cellulose crystallinity, delignification [43], and hydrolyzation of the hemicelluloses [44]. This process was widely used in the physicochemical pretreatment of lignocellulosic biomass, since there was no need to add chemicals in the process [45], basically eliminating three waste emissions, and therefore was an effective environmental friendly treatment method [46,47]. The SE was first invented by M.H. Mason in the United States in 1928, and was mainly used in the production of artificial fiberboard [48]. Initially, saturated steam from 7 to 8 MPa was used as a medium, which was not well promoted due to the high blasting pressure [48]. In the 1970s, the technology was widely used in animal feed processing, and extraction of ethanol and special chemicals from wood fiber. SE was mainly produced by an intermittent method. That is to say, after the material input, it was sequentially subjected to high temperature and high pressure, and instantaneously blasted in a closed reactor, during which time no feeding was carried out. The method for the separation of lignocellulose components was first seen in Delong’s patent [49]. After the 1980s, this technology received renewed attention and rapidly developed. Continuous SE production technology and equipment have emerged, and the application fields have been further expanded, including the fermentation of feed and ethanol [50], municipal waste and sewage, sludge processing [51], tobacco processing [52], and applications in the food and pharmaceutical fields.

Steam explosion technology, as a new green food processing technology, has attracted extensive attention through research and applications in the food field in recent years with its advantages of strong applicability, short-term high efficiency, no pollution, and industrial amplification. Biomass was heated at high temperatures between 160 and 280 °C for 10 to 30 min under high pressure (i.e., 0.69–4.83 MPa) during the steam explosion. Most of the early SE methods were of the thermal spray type and screw extrusion type [53]. Due to the long pressure relief time and low energy conversion efficiency of the above technologies, the process of “explosion” could not be reflected. In view of this, the instantaneous ejection steam explosion technology came into being. The equipment is mainly composed of three parts: steam generator, steam explosion chamber, and material receiving bin. The key innovation of the technology is the introduction of a piston valve drive system. When the holding pressure ends, the piston bursts out of the cylinder causing an instantaneous blasting motion to release the sealed state of the cavity, and ejects the material and steam together within milliseconds [43]. According to its function characteristics, the steam explosion process is mainly divided into two stages: the high temperature cooking stage and instantaneous decompression stage. In the high-temperature cooking stage, the raw material was maintained under a saturated steam of pressure and temperature for a period of time, and the hemicelluloses and other parts of the plant raw materials were hydrolyzed to form soluble carbohydrates. At the same time, the lignin in the composite intercellular layer was softened and partially degraded, which reduced the lateral bonding strength of the fibers; and the cell pores were filled with high-pressure water vapor, which became soft and plastic. During the instantaneous decompression stage, the pore gas undergoes sharp expansion due to the sudden decompression, resulting in an explosion. The fiber material was split into small fiber bundles, which resulted in modification of the physical structure and redistribution of partial components. By employing SE, it is possible to destroy the crystal structure of cellulose and make hemicelluloses easier to use in subsequent processes. In the steam explosion, there following processes were mainly contained: acid hydrolysis, thermal degradation, mechanical fracture, hydrogen bond destruction, and structural rearrangement [54,55]. Two different SE pretreatment models are shown in Figure 1 and Figure 2.

As an industrialized technology, SE has economical and eco-friendly advantages in processing large quantities of food materials [57,58]. The technology of steam explosion pretreatment has been investigated for food production from a wide range of aspects, including wall-breaking extraction of natural products, oil extraction, hydrothermal conversion of active ingredients, comprehensive utilization of raw materials and DF modification. Several studies and applications of steam explosion processing of medicinal and edible fiber show that: on the one hand, the physical explosion of steam explosion can break the multi-scale anti-extraction barrier structure at the level of the plant tissue-cell-cell wall, improving the wall-breaking and solubilizing effect of active ingredients. It has been applied to the extraction and strengthening process of various active ingredients in various raw materials, such as *Radix astragali*, *Rhus chinensis* mill [59], and wheat bran, resulting in the extraction yield and efficiency being significantly improved. On the other hand, the thermochemical action of steam explosion promotes hydrolysis reactions of the intrinsic structural components and active ingredients of plant materials. It has a deglycosylation effect on some glycoside active ingredients, which separates the sugar group from the aglycone and promotes high activity. Effective utilization of aglycone components was demomstrated’. There have been studies using steam explosion to hydrolyze rutin glycosidic bonds to prepare quercetin, process turmeric to extract and prepare diosgenin, and combine steam explosion and solid-state fermentation to convert *Polygonum cuspidatum* resveratrol glycosides to prepare resveratrol. In recent years, the technology of steam explosion pretreatment has been investigated for DF modification from a wide range of feedstocks, including wheat bran, grapefruit peel, okara, sugarcane bagasse, pineapple peel, citrus pomace, and vegetable waste [60,61,62,63,64,65,66].

## 5. Effect of SE on the Soluble Modification of DF

In accordance with the principle of SE technology, this technology is very suitable for DF modification processing. As shown in Figure 2, SE technology is commonly used for the dietary fiber modification of agricultural products such as fruits and vegetables, grains, edible fungi, and aquatic products. Its typical characteristic is that the IDF in the materials will be moderately reduced, but the SDF content increased. With the further increase of steam explosion pressure and pressure holding time, the macromolecular polysaccharides will be degraded or depolymerized to different degrees [63]. Thus, the content of DF will increase first and then decrease. The modification conditions of SE technology in grains, fruits, and vegetables and yield changes in DF are shown in Table 1. When the SE strength was 1.5 MPa for 30 s, the SDF content from okara increased to 36.28%, 26 times higher compared with the control [63]. Additionally, with appropriate steam explosion conditions, the yield of SDF in raw materials, orange peel, soybean hulls, and sweet potato residue will significantly increase [67].

Steam explosion has the characteristics of both the mechanical modification and thermal modification of fibrous materials. The structure of high-fiber raw material was modified by the effects of acid-like hydrolysis, thermal degradation, mechanical rupture, hydrogen bond rupture, and structural rearrangement generated in the SE process. On the one hand, the interlaced lignin, cellulose, and hemicelluloses, staggered distributed in lignocellulosic biomass structures, were separated to a maximum extent by SE. Through SE technology, the physical structure was destroyed, loose microstructure and honeycomb-like porous structure obtained, and the specific surface area increased, thereby improving the extraction efficiency of SDF. On the other hand, due to the destruction of the lignin, cellulose and hemicelluloses structures, the cell wall structure was disintegrated, resulting in improving the yield of SDF through thermochemical degradation that converts IDF to SDF. This is mainly achieved because steam explosion can remove a large amount of hemicelluloses from fibrous raw materials [68]. In addition, the high-pressure thermoacidic environment during the steam explosion process led to massive hydrolysis of hemicelluloses on the cell wall, and the lignin wrapped around cellulose were hydrolyzed and softened. Additionally, SE was able to readily break the hydrogen bonds of the starch structure, leading the starch to partially degrade and gelatinize; the advanced spatial structure of the protein shifted to denaturation, all of which resulted in the conversion of IDF to SDF [58]. With the increase in steam explosion strength, the conversion trends become more obvious. The results of steam explosion of lignocellulosic biomass materials show that the removal rate of hemicelluloses can reach more than 85%. The principle is shown in Figure 3 [69].

**Table 1 foods-11-03370-t001:** Effect of Steam explosion on the dietary fiber content of different food raw materials.

Material Category	Steam Explosion Conditions	DF Content Changes	Reference Documentation
Wheat bran	160 °C, 5 min	Water extractable arabinoxylan content was increased from 0.75% to 2.06%.	[70]
	0.8 MPa, 5 min	The SDF content increased to 9.62 g/100 g, which is 2.08-fold higher than that of the untreated.	[58]
Okara	1.5 MPa, 30 s	SDF content increased to 36.28%, higher by 26 times compared with the control okara.	[63]
Soybean hulls	1.2 MPa, 180 s	The SDF yield rate was 11.12%.	[67]
Sweet potato residue	0.35 MPa, 122 s	The SDF yield rate was increased from 3.81 ± 0.62% to 22.59 ± 0.35%.	[71]
*Rosa roxburghii pomace*	0.87 MPa, 97 s	The SDF content was increased from 9.31 ± 0.07% to 15.82 ± 0.31%.	[72]
Grapefruit peel	0.8 MPa, 90 s	The pectin yield rate reached 17.5%.	[62]
Orange peel	0.8 MPa, 7 min, combined with 0.8% sulfuric-acid soaking	SDF was increased from 8.04% to 33.74% in comparison with the control.	[17]
Apple pomace	0.51 MPa, 168 s, sieving mesh size, 60.	The SDF yield from apple pomace after SE was 29.85%, which is 4.76 times the yield of SDF (6.27%) in untreated apple pomace.	[73]
Okra seed	1.5 MPa, 5 min	The SDF content reached 6.5%.	[74]
*Ampelopsis grossedentata*	0.4 MPa, 4 min	The yield of crude polysaccharides reached 5.35 ± 0.12%, which is an increase of 2.2 times over that of the materials without SE pretreatment.	[75]

## 6. Effect of the SE on the Structure of DF

### 6.1. Effect of the SE on the Apparent Morphology of DF

DF is a major component of the cell wall. Steam explosion (SE) is a type of physical method used in the DF modification process in recent years. The sample material was placed into a closed environment with high temperature and high pressure, and the pores of the material filled with steam. The cell wall structure and crystal structure of cellulose were torn and destroyed by the mechanical and thermochemical action. When the high pressure is instantly released, the superheated steam in the pores will vaporize rapidly and the volume will expand sharply, leading to the “explosion” of cells. The cell walls broke out to become porous, and the chemical bond between lignin-cellulose-hemicelluloses was destroyed. Meanwhile, the lignin and cellulose structures were loosened and softened, and the hemicelluloses partially hydrolyzed, resulting in the release of low molecular weight substances in cells. In the process of SE treatment, cellulose and hemicelluloses will be degraded into SDF by acid-like hydrolysis, thermal degradation, mechanical fracture, and hydrogen bond destruction. The action principle of steam explosion is shown in Figure 4 [76].

Studies have shown that after SE treatment, the DF raw material is decomposed into fine fiber bundles, which reduces the toughness and is easier to grind. Aktas et al. found that through SE treatment, the particle size of the 50% bran was reduced from 420 to 177 μm, and the particle diameter of the 90% sample was decreased from 902 to 410 μm [70]. After SE treatment, the structure of okara significantly changed. Under weak SE strength, the layered structure of okara was destroyed into scattered small fragments, while the internal structure was relatively intact. When the strength of SE was further increased, the nucleated structure inside okara was gradually destroyed and cracked, forming small particles. Due to the destruction of the loose sheet structure on the surface of okara, the physical properties, including water and oil-holding capacities, and the swelling capacity significantly decreased. In addition, the water solubility of okara increased as the large lamellar structure cracked into small fragments and the internal nucleated structure disintegrated into small particles under high temperature and pressure [63]. Kong et al. [77] indicated that SE greatly disrupted the structure of wheat bran. The thin rod-like fibers of bran are crimped and broken, expanded into a porous structure, and a concave pore-like structure appears on the surface. This indicated that the SE seemed to have an effect on decreasing the toughness and increasing the brittleness of wheat bran. Therefore, wheat bran was easily crushed by a grinder. In another study, Sui, et al. studied the effect of SE treatment on the apparent morphology and microstructure of wheat bran. The results showed that the surface of raw bran was intact and smooth with no apparent fractures or debris. After SE, some obvious changes of bran were observed: the surface began to lose its luster and became porous, rough, and even honeycombed (as shown in Figure 5) [58].

Above all, steam explosion leads to changes in the morphology and structure of DF, which usually shows the honeycomb porous structure formed inside, the number of cracks and roughness increase, and the granulation is obvious, leading to the structure and physical properties being completely changed after the steam explosion. Additionally, with the SE strength increased, the melting of the particle surface was intensified. The specific reason is that during the high-temperature cooking process of SE, the fiber structure is softened, and some structural components are depolymerized and degraded, thereby the surface is in a molten state. Meanwhile, in the subsequent instantaneous explosion process, the liquid water in the porous structure gasified and the water vapor adiabatically expanded, which destroyed the stereoscopic structure and exposed the internal structure, resulting in increasing the complexity of the system.

### 6.2. Effects of SE on the Molecular Structure and Properties of DF

Rapid pressure release during SE treatment causes intricate mechanical action. Meanwhile, the residence time of the materials with the high-pressure steam produces various chemical reactions. The typical results of this treatment includes significant disintegration of the lignocellulosic structure, hydrolysis of the hemicellulosic fraction, and depolymerization of the lignin components, changing the molecular structure of DF and its monosaccharide composition. 

Li et al. [63] assessed the impact of steam explosion (SE) treatment on the dietary fiber, and physiochemical and protein properties of okara. The findings demonstrated that as SE strength rose, the molecular weight of the okara polysaccharide reduced and the proportion of low molecular weight fractions increased. Additionally, the molecular weight distribution range decreased. Okara polysaccharide, which has a molecular weight of 1.5 KDa, had a symmetrical peak at 2.0 MPa for 60 s and 120 s, showing that certain high molecular weight polysaccharides were gradually degraded into low molecular weight polysaccharides and oligosaccharides, increasing the amount of soluble dietary fiber. Furthermore, as SE intensity increased, so did the degree of polysaccharides breakdown and content of SDF. When the SE strength was too high, the polysaccharides were excessively degraded to convert into monosaccharides or oligosaccharides, which cannot be precipitated by 75% ethanol, leading to the decrease of contents of TDF and SDF. Another study discovered that pretreatment with SE reduced the molecular weight of *A. grossedentata* polysaccharide (AGP), implying that some high molecular weight polysaccharides were gradually degraded into low molecular weight polysaccharides and oligosaccharides, resulting in an increase in soluble dietary fiber. The results of monosaccharide composition analysis showed that AGP was composed of mannose, glucuronic acid, galacturonic acid, galactose, rhamnose, xylose, glucose, and arabinose with a molar ratio of 1.00:1.02:1.24:1.30:3.90:6.44:20.46:24.85. Meanwhile, AGP-SE had eight monosaccharide species, which was consistent with AGP. However, there was a difference in monosaccharide molar ratios between the two polysaccharides. As a result, the SE pretreatment had no discernible influence on the polysaccharide monosaccharide composition [75].

He et al. used SE to improve the physicochemical and functional properties of tartary buckwheat dietary fiber, and found that after SE pretreatment the surface structure changed from a dense intact to loose multilayer structure, and a loose multilayer honeycomb-like network structure was observed in the treated SDF. Meanwhile, the content of D-galacturonate and glucuronose in SDF significantly reduced by 40.78% and 15.07% compared with that of the untreated. However, the content of D-mannose and D-glucose were notably increased by 126.36% and 59.23%, respectively [78].

Many studies showed that the molecular weight of SDF after steam explosion treatment was reduced, and the steam explosion treatment destroyed the molecular chain of SDF. During the SE process, the hydrogen bonds and ordered structure of cellulose were broken with the high temperature and high pressure. Furthermore, the stable hydroxyl groups disordered and the macromolecular crystal was destroyed, leading to degradation of the polysaccharide from macromolecules into low molecular weight polysaccharides. Therefore, cellulose and hemicelluloses are degraded into soluble sugar such as oligosaccharides, monosaccharides, furfural, and hydroxymethylfurfural with SE treatment [79]. However, if the SE strength further increased, this soluble sugar will be further degraded into lower molecular weight carboxylic acids such as acetic acid and levulinic acid, or polymerized again [80]. Additionally, the steam explosion method causes glycosidic linkages in hemicelluloses and celluloses to hydrolyze. Furthermore, it causes lignin to depolymerize by cleaving β-O-4 ether and other acid-labile bonds. The SE technique can also release insoluble bound phenolic compounds by degrading the ester bonds between lignin and phenolic acids [81].

### 6.3. Effect of SE on the Crystallinity of DF

The crystalline state of the cellulose samples was determined by the X-ray diffraction isothermal method. Furthermore, the ICDD PDF-2 (2008) database of XRD patterns was used for analyzing the XRD patterns. According to Segal’s method, the crystallinity of each sample was determined by the intensity of the crystal peak at 2θ of ~22° and the minimum intensity of the crystal peak between 16.5 and 22.6°, corresponding to the amorphous area [82]. This method was recently used by Tanpichai et al. [83] on cellulose microfibers isolated from pineapple leaves using SE. The authors found that all treated pineapple leaf fiber samples exhibited the outstanding peaks located at 2θ of 14.8, 16.5, and 22.6°, corresponding to (101), (101¯) and (002) lattice planes of the typical cellulose I structure, respectively, while the peaks located at 2θ of 12, 20 and 22° would be determined from the cellulose II structure, which means that the crystal structure of pineapple leaf fiber after SE treatment cannot be changed. However, with the increase of the steam explosion treatment cycle, the crystallinity index increased. Similar improvement in the crystallinity of *Opuntia ficus indica* fibers treated with three green chemistry methods was found by Marwa et al. [84]. With the steam treatment assisted by lemon juice to extract cactus rackets fibers, the crystallinity was improved with the destruction of the amorphous phase of the sample and retention of the crystalline phase. In other research, the authors used SE to treated chitin and systematically investigated the crystallinity using X-ray diffraction. The results indicated that with the powerful seepage force of the steam during SE, the crystallinity index of chitin decreased 10.2% in the (110) plane and 13.3% in the (020) plane, indicating that the crystallinity of the chitin was destroyed by SE treatment [85]. Therefore, a longer time and higher pressure in SE treatment will decrease the crystallinity of the treated fibers due to the cellulose degradation.

### 6.4. Effect of SE on the Thermal Stability of DF

The thermal stability of the biomass is analyzed by using thermal gravimetric analysis (TGA), a widely adopted technique to determine the thermal degradation of plant biomass [86,87]. According to the research, the thermal degradation reaction of cellulose can be divided into four stages. The first stage, from room temperature to 160 °C involves the evaporation of extractives or water. The second stage (160–285 °C) is the thermal depolymerization of hemicelluloses or pectin. The major degradation occurred in the third stage (285–380 °C), which is attributed to cellulose degradation. The fourth stage (380–550 °C) corresponds to lignin degradation [88,89]. Liu et al. [90] used SE pretreatment to process sweet potato vine with different moisture content, and found that the hemicelluloses shoulder of manually dried feedstock samples’ derivative thermal gravimetric (DTG) curve almost disappeared after SE pretreatment, and the height of the maximum peak increased 12.5% and moved to a higher temperature due to the degradation of hemicelluloses and the related lifting of cellulose. In the study of Sui et al. [66], the authors investigated SE processing on the deconstruction of wasted broccoli stems. The results showed that the SE process led to the removal of hemicelluloses, re-condensation of lignin, degradation of the cellulosic amorphous region, and enhancement of the thermal stability of broccoli wastes. Furthermore, Tanpichai et al. [83] found similar results for thermal properties from pineapple leaves fibers after steam explosion treatment. However, Jacquet et al. [91] investigated the effect of intensity of a steam explosion on the thermal degradation of a cellulose. The results indicated that the thermal stability of the steam explosion samples decreased with the intensity of the treatment, which was contrary to the above findings. This phenomenon may be because of the high intensity of SE and the glucose ring cracking reaction. Thus, SE treatment has an obvious impact on the thermal stability of DF.

## 7. Effect of the SE on the Physicochemical Properties of DF

The capacity of DF to interact with water via physical adsorption is a significant aspect of its physiological activity [72]. The results of swelling capacity can be used to determine how much water and volume DF can hold [92]. Through changing the structure of DF raw materials, SE treatment increases the content of SDF and soluble small molecular substances, reducing IDF. Moreover, the microstructure of the material changes from dense to lose and porous, resulting in changes in the functional properties of DF, such as WHC, SC, OHC, and other physicochemical activities. Based on the modification of the conditions of steam explosion, the property changes of DF are shown in Table 2.

In much research, SE pretreatment can effectively improve the WHC and water-retention capacity (WRC) of DF. Moderate steam explosion treatments can increase the water-retention capacity coupled to an increase of the overall crystallinity [93]. In the study of Zhao et al. [61], the outcomes demonstrate that SE pretreatment can raise the WHC of wheat bran. The increase in soluble dietary fiber causes a decrease in the WRC of wheat bran, which could be ascribed to the exposure of hydroxyl and carboxyl groups bound to water groups by the instantaneous shear stress of SE. In addition, Zhai et al. investigated the effect of SE technology on the modification of DF from *Rosa roxburghii pomace*. The results revealed that the physicochemical properties of SDF and IDF were improved after SE treatment. Compared with without SE treatment, the hydration capacity of IDF was significantly increased from 6.77 ± 0.33 g/g to 7.92 ± 0.27 g/g. This is possibly because the molecular morphology of SE-IDF was changed as the specific surface area and rougher surface structure of the material increased, exposing more binding sites for water molecules and so enhancing the ability of DF to capture water molecules [72].

Oil-holding capacity is one of the important indexes of physical and chemical properties of DF [30]. Research indicates that the OHC mainly relates to the protein structure and absorption property. SE treatment caused the depolymerization of proteins, which might lead to the decrease of oil holding capacity. A similar result was observed in Li’s study [63], that is, the oil-holding capacity of okara decreased with the increase of SE strength. At SE strengths of 2.0 MPa for 120 s, the oil-holding capacity decreased 60.9%, compared with the control. In the study of Zhai et al. [72], the results showed that the OHC of the modified IDF was 1.3 times higher than that of IDF before modification by SE, and the OHC of SDF increased from 1.53 ± 0.51 g/g (0-SDF) to 5.96 ± 0.49 g/g (SE-SDF) [94]. It is possible that SE treatment led to the porous structure and large specific surface area, which are beneficial for the absorption property of dietary fiber.

Dietary fiber is a general term for non-starch polysaccharide and lignin. Due to the molecular structure, non-starch polysaccharide contains many hydrophilic groups, which play the role of hydrophilic hydroxyl groups, and has certain hydrophilic properties and exhibits certain swelling capacity (SC). Wang et al. [71] found that SE treatment increased SC values of DF from sweet potato by 37.79%. In another study, the authors used SE treatment conditions to optimize SDF extraction from apple pomace [73]; the results illustrated that the SC was increased, which may be attributed to the increase in the surface area of SDF after SE. Additionally, it was applied by Wang et al. to prepare SDF from orange peel assisted by steam explosion and dilute acid soaking, and found that the SC of SDF from orange peel treated by SE-SAS increased from 4.83 ± 0.52 mL/g db to 6.28 ± 0.73 mL/g db. The SC was dependent on the characteristics of individual components and the physical structure (porosity and crystallinity) of the fiber matrix [95]; thereby, the increase in SC might be attributed to a rise in the amount of short-chains and surface area of dietary fiber induced by SE.

## 8. Effect of SE on the Functional Activities of DF

SE treatment not only changed the structural composition of DF, but also improved the biological activity of DF. After SE treatment, the specific surface area of DF was increased, and more groups were exposed. Meanwhile, a part of the hydrogen bonds inside the cellulose were broken to form free hydroxyl groups. The uronic acid content in SDF was significantly increased after SE treatment, which had a positive effect on the adsorption of excess cholesterol and toxic cations [96].

Wang et al. [17] found that the binding capacity of SDF from orange peel treated by SE–SAS to three toxic cations (Pb, As and Cu) was significantly improved. Furthermore, compared with the control group, the main weight-average molecular weight of SDF in orange peel treated by SE-SAS was obviously smaller and the thermal stability of SDF was higher. In another study, the authors illustrated that SE can increase the cation exchange capacity (CEC) of *Polygonatum odoratum* DF [97]. Three possible mechanisms were proposed for the binding capacity of dietary fibers to heavy metals: chemical adsorption, physical adsorption, and mechanical adsorption [98]. Chemical adsorption is related to the existence of ophenolic groups in lignin and carboxyl groups in uronic acid. When the pH value was increased, the carboxyl groups were dissociated into carboxyl anions (RCOO·), which showed stronger interaction with the toxic cations, resulting in higher binding capacity of the dietary fibers with the cations. Physical adsorption was caused by van der Waals’ force, which was connected with temperature, while mechanical adsorption depended on the degree of porosity of DF and its ability to trap the substances in its spatial structure.

According to the research, it is shown that the bran constituent of cereal is rich in phenolic compounds, which has a positive effect on oxidative stress. Phenolic acids can be divided into three types: free, soluble esters, or conjugates and insoluble combined forms. Insoluble bound phenolics are abundant and combined with ester and ether bonds of cell wall components such as arabinoxylans and lignin, showing significantly higher antioxidant capacity compared with soluble free and conjugated phenolics [99,100,101]. The bound phenolic acids (BPA) can be released by steam explosion-assisted extraction. Meanwhile, the content of phenolic acids and its antioxidant properties were significantly improved [102]. Kong et al. [103] used SE treatment to process wheat bran; the results showed that the phenolic content and total flavonoids were increased by 83% and 198%, respectively, at SE strengths of 0.8 MPa for 5 min. Li et al. [104] found that with SE treatment of 1.5 MPa for 60 s, the amount of free and bound phenolics in the tartary buckwheat bran were increased from 23.62 and 0.36 mg gallic acid per gram dry weight (mg GAE per g DW) of tartary buckwheat bran to 27.34 and 0.99 mg GAE per g DW, which increased by 15.7% and 175%, respectively. Moreover, the biological activity tests indicated that SE effectively increased the oxygen radical absorbance capacity (ORAC) in vitro of the bound phenolics by 270%. It also enhanced the cellular antioxidant activity (CAA) in vitro of free phenolics by 215%. In another study, after SE treatment at 220 °C for 60 s, the yields of free and conjugate ferulic acid increased by about 59.0 and 8.45 times, respectively. Meanwhile, the corresponding increases of p-coumaric acid were 47.6 and 7.25 times. With the residence time prolonged to 120 s, the total soluble phenolic content from barley bran reached 1686.4 gallic acid equivalents mg/100 g dry weight, which was about 9.83 times higher than that of the untreated sample. Therefore, the extraction rate of free phenolic acids and soluble conjugates increased sharply after steam explosion, and some bound phenolics like ferulic acid and p-coumaric acid in the cell walls might be released. 2,2-Azinobis (3-ethylbenzothiazoline-6-sulfonic acid (ABTS) and ferric-reducing antioxidant power (FRAP) assays indicated that SE can obviously improve the antioxidant capacity of the Tibetan hull-less barley bran extract [81].

In addition, DF possess several biological properties such as hypoglycemic activity, cholesterol-lowering activity, and prebiotic activity. The results obtained from Chen et al. [105] indicated that the content of SDF was increased from 2.6 ± 0.3% to 30.1 ± 0.6% from soybean residues treated by SE combined with extrusion treatment (BEP). Moreover, in vivo experiment results indicated that after BEP processing, the SDF significantly reduces the concentrations of total cholesterol (TC), low-density lipoprotein cholesterol (LDL-C), and triglyceride (TG) while increasing the concentration of high-density lipoprotein cholesterol (HDL-C). In other research, Liu et al. [75] observed that SE pretreatment can increase the content of polysaccharide from *Ampelopsis grossedentata* and slightly change the physicochemical properties of AGP (i.e., molecular weight, monosaccharide composition, and glycoside bonding). Furthermore, the polysaccharides pretreated with SE showed significantly higher α-glucosidase inhibitory activities than those from the without SE-pretreatment sample, which might serve as an efficient α-glucosidase inhibitor. Additionally, SE treatment can obtain fiber components with potential prebiotic activity and high antioxidant activity from asparagus byproducts [106].

## 9. Conclusions

SE technology has been widely used in the treatment of lignocellulosic materials. With the development of equipment and technology, it has been gradually applied in the food industry, pharmaceutical industry, biological energy, chemical materials, environmental protection, and other fields. This technology has its significant advantages in the processing of DF materials, but also has many disadvantages. 

### 9.1. Analysis of Technical Advantages and Disadvantages

As shown in Table 3, SE treatment has obvious technical advantages, but also has defects. SE technology is widely used in the modification of DF because of its low cost, simple operation only needing to pass high temperature and high pressure steam, no chemical additives and low pollution of the environment [107]. Meanwhile, SE treatment can destroy the connection of cellulose, promote the release of functional components in materials, and also modify DF to enhance the corresponding functional properties. With short time consumption, large-scale production, and relatively low energy consumption, SE is capable of improving the comprehensive utilization and added value of materials.

However, there are also some deficiencies in SE treatment. Due to its complex physical and chemical processes, it is difficult to accurately control the strength and uniformity of treatment, which makes it easier to cause the degradation of other effective components and occurrence of Maillard reaction, and the treatment cannot be continuous. 

### 9.2. Development Trends

(1)Further study the change process and mechanism of DF during SE treatment, and further clarify the exact effect of SE treatment on food safety and its influence on nutritional value;(2)Further clarify the physical and chemical changes in DF materials during SE treatment, to clarify the structure-activity relationship between its chemical components and functional activities, and reveal the molecular mechanism of the biological activity change in DF materials in SE treatment;(3)Further improve the research and upgrade of industrialized SE processing equipment and facilities; develop in the direction of large-scale, continuous, automatic and precise control; and explore combined process technologies that combine multiple pretreatment methods.

SE treatment technology is an efficient modification technology of fiber material, which has the advantages of low investment, low energy consumption, short time consumption, and environmental friendliness. However, the current research mainly focuses on the industrial application of wood fibers. Limited information is available on the modification of DF in food by SE treatment, mainly in the grain bran, and fruit and vegetable byproducts. After SE treatment, under the dual action of high temperature and high-pressure steam, the physical and chemical properties of the material significantly changed. After process optimization, the physical and chemical quality and nutritional value of DF have been markedly improved. Therefore, finding a suitable method that is more conducive to industrialization and improves the yield and functional properties of DF will continue to be a hot topic in the food industry.

## Figures and Tables

**Figure 1 foods-11-03370-f001:**
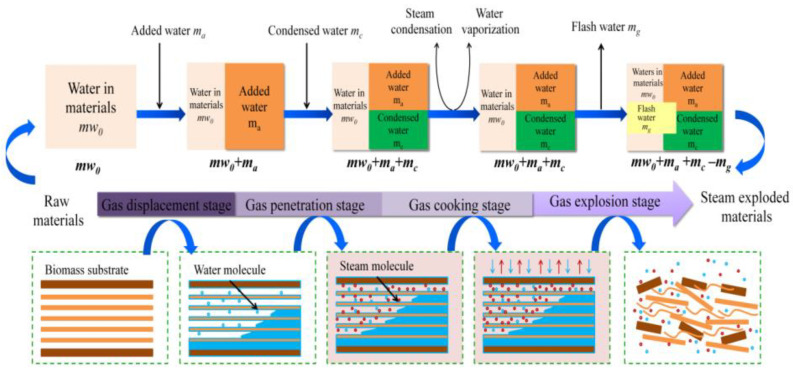
Schematic diagram of steam explosion process [56]. Reproduced with permission from Wenjie Sui and Hongzhang, Industrial Crops and Products; published by Elsevier, 2015.

**Figure 2 foods-11-03370-f002:**
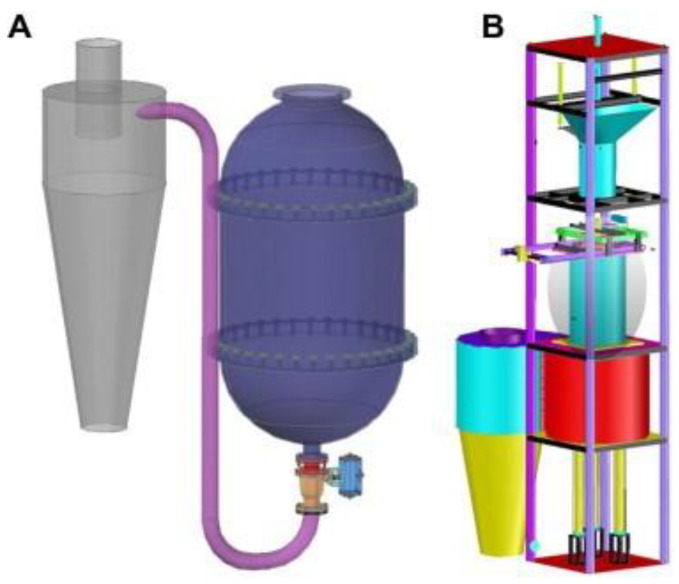
The structure diagram of two SE pretreatment models. (**A**) A model in valve blow mode. (**B**) A model in catapult explosion mode [43]. Reproduced with permission from Zhengdao Yu, Bailiang Zhang, Fuqiang Yu, Guizhuan Xu and Andong Song, Bioresource Technology; published by Elsevier, 2012.

**Figure 3 foods-11-03370-f003:**
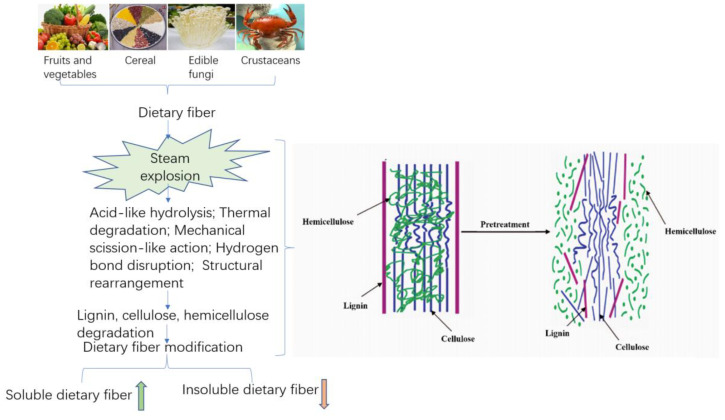
Schematic diagram of the modification mechanism of dietary fiber in Steam explosion.

**Figure 4 foods-11-03370-f004:**
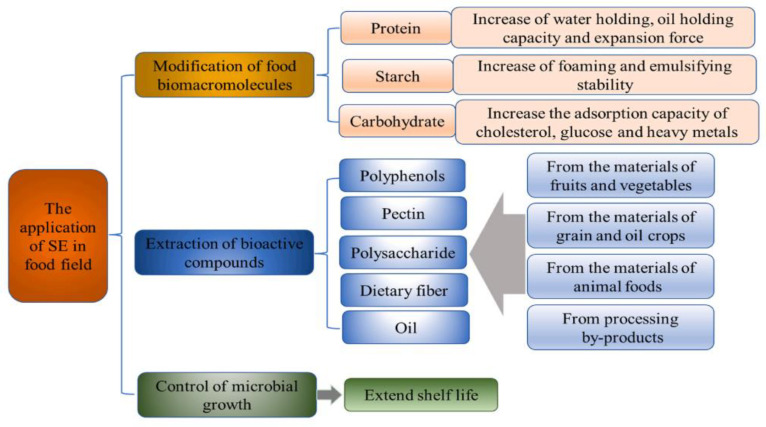
Schematic diagram of action principle of steam explosion [76]. Reproduced with permission from Fachun Wan, Chengfeng Feng, Kaiyun Luo, Wenyu Cui, Zhihui Xia and Anwei Cheng, Trends in Food Science & Technology; published by Elsevier, 2022.

**Figure 5 foods-11-03370-f005:**
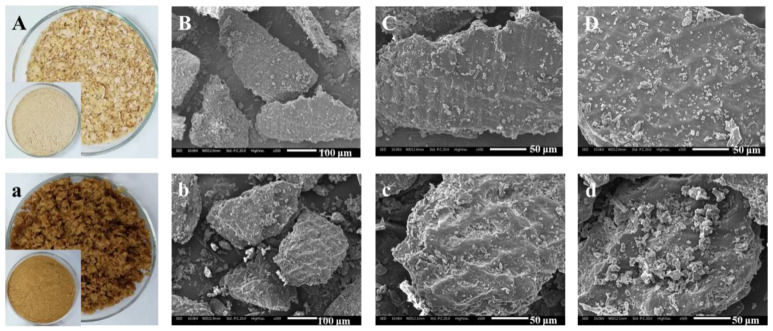
Photographs and SEM micrographs of raw bran and SE bran. (**A**,**a**): photographs of raw bran and SE bran; (**B**–**D**): SEM micrographs of raw bran powders through a 60 mesh screen (**B**): 200×; (**C**,**D**): 500×; (**b**–**d**): SEM micrographs of SE bran powders through a 60 mesh screen (**b**): 200×; (**c**,**d**): 500×.

**Table 2 foods-11-03370-t002:** Effect of Steam explosion on the physicochemical properties of dietary fiber.

Material	Modification Conditions	Property Changes	Reference Documentation
Wheat bran	0.1~1.5 MPa, 5 min	WHC↑, WRC↓, cellulose content↓	[70]
	0.8 MPa, 5 min	SDF content↑, IDF↓, WHC↑, SC↑, OHC↑	[70]
Okara	1.5 MPa, 30 s	SDF content↑, WHC↓, SC↓, OHC↓	[63]
Sweet potato residue	0.35 MPa, 122 s	SDF content↑, WHC↑, SC↑, OHC↑	[71]
*Rosa roxburghii pomace*	0.87 MPa, 97 s	SDF content↑, IDF↓, WHC↑, SC↑, OHC↑	[72]
Orange peel	0.8 MPa, 7 min, combined with 0.8% sulfuric-acid soaking	SDF content↑, WS↑, WHC↑, SC↑, OHC↑, emulsion stability↑	[17]
Apple pomace	0.51 MPa, 168 s, sieving mesh size, 60	SDF content↑, WHC↑, SC↑, OHC↑	[73]
Cellulose fibers	1.0~4.0 MPa, 5 min	crystallinity↑, Water retention values↑	[74]

“↑” means this characteristic was improved after SE treatment. “↓” means this characteristic was reduced after SE treatment.

**Table 3 foods-11-03370-t003:** Comparison of the advantages and disadvantages of steam explosion and conventional treatment methods.

Subjects	Advantages	Disadvantages	Reference
Industrial application cost	Short time consumption, Large-scale production, low cost	Equipment cost investment is large in the early stage.	[7]
Operation convenience	Higher automation and lower operation intensity	Continuous production cannot be realized.	
Operation safety	Better equipment safety protection, less potential safety hazard in operation	The equipment requires high temperature and high pressure, which will generate huge noises during work. It needs a muffler device, and an exhaust gas absorption or waste heat recovery device.	
Product safety	Except high-pressure steam, no other chemicals are brought into the process, which will not pollute the materials.	High temperature and high-pressure process is prone to adverse reactions such as Maillard reaction and denaturation of nutrients.	

## Data Availability

Not applicable.

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
