# Peer review of "Principle and Application of Steam Explosion Technology in Modification of Food Fiber"

_foods, 2022, doi:10.3390/foods11213370_

Round 1
Reviewer 1 Report
Manuscript ID: foods-1957118: Principle and Application of Steam explosion Technology in Modification of Food Fiber
1. Line 45: the word „Women“ should be written in lower case;
2. Lines 49-52: should insert some literature for this;
3. Line 83: Delete „;“;
4. Line 94: state which was the first definition when it is already mentioned;
5. Line 126: you have "cellulose" written twice;
6. Line 142: the word „Considering“ should be written in lower case;
7. Lines 142-149: should insert some literature for this;
8. Lines 153-160: this paragraph is very similar to what was already said in the introduction (Lines 58-69) and the same literature is cited, there is no need to repeat it;
9. Line 209: the word „When“ should be written in upper case;
10. Line 234: the word „Wall-breaking“ should be written in lower case;
11. Line 242: the word „On“ should be written in lower case;
12. Line 272: delete „which were“;
13. Line 273: change „were separated maximum extent“ into „“were separated to a maximum extent“;
14. Table 1: change symbol „、“ into „comma - , “;
15. Figure 2: in the figure name, the words „Dietary Fiber“ should be written in lowercase letters;
16. Line 297: change „was“ into „were“;
17. Lines 294-298: this paragraph is very similar to what was already said, there is no need to repeat it;
18. Line 310: you should explain what the labels you use here mean ((d50) (d50) and d90). Also, if you put (d50) in parentheses, put a d90.
19. Line 321: change „Which“ into „This“;
20. Line 322: change „had the effect“ into „to have an effect“;
21. Line 327: „particleization“ – did you mean “particularization“?;
22. Line 334: the word „Stereoscopic“ should be written in lower case;
23. Figure 3: in the figure name, you should explain what the Abbreviations you use here first time mean (NWB and SEWB);
24. Line 397: the word „On“ should be written in lower case;
25. Line 417: you should explain what the Abbreviation you use here first time means (TGA);
26. Line 426: you should explain what the Abbreviation you use here first time means (DTG);
27. Line 465: „Many researches“ - you should specify some;
28. Lines 464-474: is there any explanation why with authors like Li the specific surface area decreases (and therefore OHC), while with some others, like Zhai, it increases with SE treatment?
29. Line 474: the word „Fiber“ should be written in lower case;
30. Table 2: The same comment as in Table 1.
31. Lines 492: there is research where the specific surface area is decreasing;
32. Line 494: the word „The“ should be written in upper case;
33. Line 512: the word „It“ should be written in lower case;
34. Line 532: the words „Barley Bran“ should be written in lower case;
35. Line 537: change „improved“ into „improve“;
36. Lines 569-572: are there any data on the amount of Maillard reaction products that are produced by the SE treatment?;
Reviewer 2 Report
Introduction
Line 56-76 - you write about different methods of DF modification. I can’t there find any information about disadvantages of physical methods and about disadvantages and advantages of chemical and biological methods.
Line 76-86 - you describe steam explosion. I can't find any information in your manuscript about use this methods in „big” scale. Can you write a few examples.
Source and classification of dietary fiber in food
Line 126 you write twice „cellulose”
Line 127 please add 1,4 …. glucose linked
Modification methods of DF
Paragraph have a lot in common information with line 60-75 from Introduction part
Steam explosion technology process and principle
What about of disadvantages of this method, because SE create a lot of new chemical compounds like inhibitors
Effect of the SE on the structure of DF
Line 384 – carboxymethylfurfural, what about hydroxymethylfurfural
Line 419-423 Hemicelluloses are degrade from about 160C, not all extractives substances are volatile. Some of them need to be dissolve if you want to remove them from material, as tannins. Oils have higher temperaturę of boiling then 180C
9. Discussion
Line 564 - You write „No toxic waste”. During SE create many organic substances. Some of them are not very healthy
Obvious
Change the word hemicellulose to hemicelluloses (with „s” in the end). Hemicelluloses are a group of polysaccharides in opposite to cellulose or lignin.
There is also lack of information about how many (in percentage) are DF in typical plant using in food industry. Which of the plant have the higher chance to change their ceel wall during SE and use productc in industry
Reviewer 3 Report
It is very important to add a reference on lines 56-58.
In line 62 change the term to melt, in addition to placing in parentheses that it is a glycosidic bond.
Please change the prhase "not a polysaccharide" to is not a polysaccharide in line 129.
In the ilne 136 is better to write Goñi et al, 2009,
write references on lines 145 and 149.
On line 153 add the references,
Add a reference on line 166
Add a reference 177 and another in line 184.
It is recommended to add to the figure 1 or 2 reactions with structures or drawings of the short processes for a better understanding of the reader.
Write in parentheses that it is a glycosidic bond on line 248
In line 250 explain what resveratrol is and why it is important.
Remove the word viz in all document.
Explain more clearly the written phrase of lines 280 to 285.
It is highly recommended to use flowcharts in the sections that warrant it, for example in paragraph 1 of section 6.1. In the same way, between lines 347 to 350. Mention the methodology used to calculate the molecular weight distribution In the same part. The same between lines 360 to 365.
In line 309 you must write by which methodology the particle size was obtained.
Do not capitalize the word Porous on line 321.
It is advisable to take a photomicrograph at higher magnifications in order to reaffirm assertions regarding porosity. In the presented micrographs it cannot be clearly observed.
Homogenize the way of writing the references, for example, KONG F et al and another says Li et al.
It is important to be more specific in line 395 regarding the 2 theta position of the compounds. Furthermore it should be mentioned that this is, according to a median identification of ICDD or CCD charts that are integrated into the diffractometer database.
Write the meaning of the acronym TGA and DTG in section 6.4
In this same section how the swelling capacity is obtained.
On line 465 it is an option to put some of the references instead of writing a lot of research.
Be very clear when write about specific surface area with respect to the method in question. Line 483.
Clarify what the acronym GAE., DW, FRAP, ABTS means.
In table 4.3 in the disadvantages section, clarify why the Maillard reaction is not good.
Reviewer 4 Report
Remarks
*Line 83, there is an extra space before the " ; "
*Line 142, the letter c in lower case "considering".
*Fig. 3, page 8. It is important to include at the bottom of the figure, the meaning of SEM (Scanning electron microscopy ?), NWB, SEWB.
*Table 3, It is recommended to include the references used, which allowed to estrablich the advantages and disadvantages of the method. Add another column.
*Discussion section.
It looks like a summary of the whole document content. I do not think this section should be called discusson, perhaps handle it as highlights, relevant aspects or summary.
